# The Modification of the Illumina^®^ CovidSeq™ Workflow for RSV Genomic Surveillance: The Genetic Variability of RSV during the 2022–2023 Season in Northwest Spain

**DOI:** 10.3390/ijms242216055

**Published:** 2023-11-07

**Authors:** Carlos Davina-Nunez, Sonia Perez-Castro, Jorge Julio Cabrera-Alvargonzalez, Jhon Montano-Barrientos, Montse Godoy-Diz, Benito Regueiro

**Affiliations:** 1Microbiology and Infectology Research Group, Galicia Sur Health Research Institute (IIS Galicia Sur), 36312 Vigo, Spain; carlos.davina@iisgaliciasur.es (C.D.-N.); jorge.julio.cabrera.alvargonzalez@sergas.es (J.J.C.-A.); benito.regueiro@usc.es (B.R.); 2Faculty of Biology, Universidade de Vigo, 36310 Vigo, Spain; 3Microbiology Department, Complexo Hospitalario Universitario de Vigo (CHUVI), Servizo Galego de Saúde (SERGAS), 36214 Vigo, Spain; bitalio.jhon.montano.barrientos@sergas.es (J.M.-B.); montserrat.godoy.diz@sergas.es (M.G.-D.)

**Keywords:** next-generation sequencing, viral enrichment, RSV, nirsevimab

## Abstract

There is growing interest in the molecular surveillance of the Respiratory Syncytial Virus and the monitorization of emerging mutations that could impair the efficacy of antiviral prophylaxis and treatments. A simple, scalable protocol for viral nucleic acid enrichment could improve the surveillance of RSV. We developed a protocol for RSV-A and B amplification based on the Illumina CovidSeq workflow using an RSV primer panel. A total of 135 viral genomes were sequenced from nasopharyngeal samples through the optimization steps of this panel, while an additional 15 samples were used to test the final version. Full coverage of the G gene and over 95% of the coverage of the F gene, the target of the available RSV antivirals or monoclonal antibodies, were obtained. The F:K68N mutation, associated with decreased nirsevimab activity, was detected in our facility. Additionally, phylogenetic analysis showed several sublineages in the 2022–2023 influenza season in Europe. Our protocol allows for a simple and scalable simultaneous amplification of the RSV-A and B whole genome, increasing the yield of RSV sequencing and reducing costs. Its application would allow the world to be ready for the detection of arising mutations in relation to the widespread use of nirsevimab for RSV prevention.

## 1. Introduction

Human Respiratory Syncytial Virus (RSV) is one of the main causes of hospitalization for children under five and for those over seventy-five. With an estimated 59,600 total deaths per year worldwide, it is also the second leading cause of infant mortality [1,2].

RSV is a single-strand negative-sense RNA virus with a genome of 15.2 kb from the family Pneumoviridae. This family includes viruses infecting the upper respiratory tract in mammals and birds. The human RSV genome codifies 10 ORFs and 11 proteins, of which the most studied ones are the attachment glycoprotein (G) and the fusion protein (F), which are both associated with the initial phases of infection. Two main variants (A and B) with several genotypes and subgenotypes circulate simultaneously [3,4,5]. According to the classification proposed by Goya et al., based on the phylogeny of the G protein, three genotypes have been defined for RSV-A and seven for RSV-B, as well as several subgenotypes [6,7]. However, since 2020, all samples published in the GISAID database belong to genotype two for RSV-A and genotype five for RSV-B. In RSV-A, over 98% of samples belong to subgenotype 2.3.5, which includes the G protein insertion 261–284. In RSV-B, all samples belong to subgenotype 5.0.5a, with insertion 245–264 in the G protein.

RSV classification is commonly based on the G protein due to its variability. Nonetheless, other areas of the RSV genome, although less variable, are also relevant for surveillance purposes. The F protein is the target of the monoclonal antibodies inhibiting viral cell entry such as nirsevimab and palivizumab and most vaccine candidates. Nirsevimab is an inhibitor of the F protein in its prefusion stage, showing a >50-fold higher potency and 5–8 times higher half life in comparison to palivizumab. Nirsevimab prescription for children has been approved in Northwest Spain for the 2023–2024 season. The F gene should therefore be checked for mutations that could influence the effectiveness of prophylaxis and treatment drugs [8,9,10].

As next-generation sequencing (NGS) has become more available and affordable, whole-genome sequencing (WGS) has become a common tool in epidemiological surveillance. Metagenomic NGS (mNGS) is possible, but yields are low due to the high amount of host nucleic acids in relation to viral nucleic acids [11,12]. Obtaining full genome coverage using mNGS requires a very high sequencing depth, which dramatically increases sequencing costs.

Viral nucleic acid enrichment involves the enrichment of the viral fraction via either the removal of non-viral nucleic acids (hybrid capture) or the specific amplification of target nucleic acids (PCR) [12]. This process makes high-coverage sequencing possible even in low-viral-load samples. While hybrid capture is a commonly used method of viral fraction enrichment, it is also a more complex and time-consuming method than PCR enrichment and, therefore, PCR-based methods are more feasible for establishing global viral surveillance [13]. Additionally, PCR-based methods have been shown to perform better than hybrid capture with low viral load samples [14]. Published protocols for PCR-based enrichment rely on the amplification of several amplicons in independent reactions, requiring between five and ten PCR reactions per sample, increasing complexity and cost, especially when working with a high number of samples [3,15,16]. A recent protocol achieved the amplification of the RSV genome in two RT-PCR reactions by multiplexing all primers from non-consecutive amplicons (odd amplicons in one pool, even amplicons in another) [17]. Using only two PCR reactions per sample causes a significant reduction in cost and, when combined with the automatization of enrichment and library preparation, allows for the high-throughput sequencing of viral genomes.

In this study, we adapted an already available primer panel for the amplification of the whole genome of RSV to a widely used workflow (Illumina CovidSeq) and to the genetic characteristics of RSV currently circulating in Europe. Our protocol amplifies both RSV-A and RSV-B in parallel, thus eliminating the need for prior subtyping and allowing for the amplification of RSV A/B coinfection samples. Additionally, a bioinformatic pipeline was developed to speed up the genotype assignment. The epidemiology of RSV in Europe was also studied.

## 2. Results

### 2.1. Development of Primers and Protocol for RSV Amplification

Firstly, the optimization of the multiplexing of a previously published primer panel [16] was performed. Based on experimental data, primers with off-target binding areas that could impact amplification were replaced. Secondly, primers were modified to account for common mutations in the last epidemiological period (October 2022–March 2023) that could alter primer binding efficiency. Finally, primer relative concentrations were modified to increase the coverage of poorly amplified sections of the genome (Appendix A).

### 2.2. Mutations in Primer Hybridization Areas Increase Complexity for Viral RNA Amplification

RNA viruses have a high mutation rate [18], and, therefore, one challenge of primer panels for whole genome amplification is the appearance of mutations in the primer hybridization areas, as these mutations could decrease primer–target binding. From the original panel used, we searched for common mutations circulating globally that overlapped with the primer binding areas. Four mutations were detected, all of them synonymous (Table 1). The oligo panel was modified with the intention of correcting this difference in binding.

Out of the four mutations detected, three belonged to RSV-A. More genetic variability across samples makes amplification more challenging, as it is more likely that a subset of samples present a mutation in the primer binding areas. It also means that more degenerate primers may be required to cover all variants present. In order to quantify the genetic variability of RSV-A and B in the last season, we calculated the pairwise genetic distance for all A and B samples during the 2022–2023 season in Europe (GISAID database). We found that genetic variability was higher in RSV-A than in RSV-B (*p* < 0.0001) (Appendix A).

### 2.3. Protocol Could Successfully Amplify Samples Circulating in Northern Spain in the 2022–2023 Period

Using the final optimized version of our primer panel, 16 samples (8 RSV-A, 7 RSV-B, and a negative control) were sequenced. Agilent bioanalyzer samples showed the expected peaks of around 2 kb for each of the pools and subtypes (Figure 1).

Sequencing was successful, showing for the 15 samples a coverage (base reads > 10) of over 85% for the whole genome, of 100% for the G gene, and of >95% for the F gene. Whole genome coverage was similar for both genotypes A and B (*p* = 0.71) (Figure 2a). The viral load (quantified as CT value in RSV RT-PCR) had an impact in the median base depth but not in the number of unread bases (Figure 2b,c).

Of eight RSV-A samples, six belonged to a cluster with the G13793A mutation. Base 13793 is located in the binding area of primer A10f (Table 1), corresponding to the amplicon 10 that covers the 5′ terminal region of the genome. Despite the use of a degenerate oligo to account for this mutation, there was a noticeable drop in coverage depth in amplicon 10 in the samples including G13793A, while in the two samples without this mutation the coverage for amplicon 10 was similar to the rest of the genome (Figure 3a). RSV-B also presented depth variability across amplicons, although without cluster-specific changes in coverage (Figure 3b).

### 2.4. Circulation of RSV in Europe during the 2022–2023 Surveillance Season

During the whole epidemiologic period and the optimization rounds of this protocol, a total of 150 RSV sequences (147 patients, three coinfections) were successfully sequenced in our facility (98 RSV-A and 52 RSV-B). They accounted for 20.4% of the total PCR-positive RSV cases diagnosed in the Vigo area during the study period. In total, 84.4% of the cases belonged to patients under 10 y/o and 80.2% were detected in the pediatric emergency department (Table 2, Appendix A).

Phylogenetic analysis was performed on the sequences obtained in this study, as well as on the other European samples from the same period (October 2022 to March 2023), and uploaded to GISAID. Two clusters were identified. The biggest one was characterized by the mutation G:P206Q and accounted for 82 of the 98 Vigo RSV-A samples. The other 16 samples corresponded to a cluster showing the mutation G:T113I. From the European GISAID sequences, 33% belonged to the G:P206Q cluster, 41% to the G:T113I cluster, and 26% to a third G:L142S cluster (Figure 4).

All RSV-B samples from GISAID in Europe during this season belonged to Goya clade B-5.0.5a, with most samples both in Europe (90.0%) and in Vigo (49/52; 94.3%) in a cluster defined by mutation G:S100G (Figure 4).

### 2.5. Detection of Mutations Associated with Resistance to Monoclonal Antibodies

We searched through our sequences for mutations associated with decreased activity with either palivizumab or nirsevimab, and monoclonal antibodies authorized for RSV treatment. RSV-A mutation F:N276A, associated with palivizumab activity decrease [19], was found in 78.6% of all the sequences studied. 

In the case of nirsevimab, two sequences included mutation F:K68N, associated with a 4-fold decrease in activity in vitro [20]. The phylogenetic analysis suggested the appearance of the mutation in a single event, as both samples were closely related. One of the samples was detected in England while the other was sequenced in our facility, suggesting that this mutation could be more extended than what is currently being detected.

## 3. Discussion

Protocols for viral genome enrichment using a PCR allow for the quick and cost-effective amplification of viral RNA and, therefore, could help sequencing facilities in the implementation of RSV WGS. Several protocols for this purpose already exist that are able to amplify either RSV-A, RSV-B, or both [3,16,21], including a protocol that, such as this one, allows for the amplification of RSV-A and RSV-B in two single RT-PCR reactions [17]. 

Our protocol, modified from a previously published primer panel [16], presents the added benefit of being adapted to the Illumina CovidSeq reagents, which is already common in sequencing facilities for performing the whole-genome sequencing of SARS-CoV-2. Additionally, our primer panel works simultaneously for RSV-A and RSV-B, so there is no need for subtyping the samples prior to the beginning of the RT-PCR. Finally, the primer panel was adapted to cover variants circulating in Europe in the last epidemiological season. All these characteristics make this protocol easily adaptable and accessible to surveillance facilities across the globe. As a final advantage, this protocol could potentially be adapted to other respiratory viruses as long as different primer panels are added and the melting temperature for the PCR is optimized.

The bioinformatic analysis of RSV sequences requires skilled specialists or a user-friendly pipeline. As our protocol does not require subtyping in RSV-A or B prior to sequencing, it requires the alignment of each read appropriately to either RSV-A or RSV-B. Our bioinformatic pipeline aligns with references for RSV-A and RSV-B in order to subtype and detect coinfections through the alignment of reads in areas of high genetic variability. 

Our optimized protocol achieved the full coverage of the G gene and over 95% of the F gene in all tested samples using the primer panel presented in this work. The G gene has been commonly used for RSV genotyping due to the high variability of its second hypervariable region, which allows for the characterization of variants [22]. The F gene is clinically relevant as it is the target for monoclonal antibodies such as palivizumab and nirsevimab [10,23]. Mutations associated with a decrease in nirsevimab activity have been detected in vitro, such as F:N67I and F:N208Y in RSV-A or F:K68N, F:N201S, and F:N208S in RSV-B [20]. We detected two sequences of RSV-B presenting F:K68N in Europe in the last epidemiological period, one in our facility and one in England, suggesting the possibility of a bigger cluster going undetected given the geographical distance between both location sites. Mutation F:K68N presented a low decrease in nirsevimab activity (4-fold), but when combined with F:N201S or F:N208S, activity reduction increased to 13,000- and >90,000-fold, respectively [20]. An increase in sequencing capacity would allow vigilance systems to find the true prevalence of F:K68N as well as to increase the vigilance of the arisal of F:K68;N201S or F:K68N;N208S variants. 

The main challenges of viral genome PCR amplification are usually off-target alignment areas for the primers and the appearance of mutations in the binding area of the primers. While we did not detect any significant off-target binding areas, point mutations in the primer hybridization areas were found (Table 1). Even after modifying the primers to their degenerated version to account for the mutations detected, these mutations caused a decrease in read number for the affected amplicons. Further optimizations are underway to increase amplicon read homogeneity. In the case of this epidemiological season, RSV-A had more genetic variability than RSV-B, with samples being divided into three main clusters. This required the use of more degenerate primers for RSV-A amplification in the panel, increasing the complexity. Future plans could include the adaptations of the panel each year by tracking mutations through the epidemiological seasons in the northern and southern hemispheres, although more sequences deposited in public repositories would be needed for this approach to succeed.

In this study, we presented the characterization of the main variants of RSV circulating in Europe during the 2022–2023 season. Samples belonged to genotypes 2.3.5 (RSV-A) and 5.0.5a (RSV-B), as defined by Goya et al. [6]. These groups included insertions in the G gene, originally detected and referred to as variants ON1 and BA for RSV-A and B, respectively [24,25]. In the years before the SARS-CoV-2 pandemic, these variants had become dominant, as has been reported in molecular characterization studies in different locations [26,27,28,29]. We have additionally separated the RSV-A samples of this epidemiological season in Europe into three distinct clusters, defined by G protein mutations P206Q, T113I, and L142S. RSV-B showed lower genetic diversity, with most samples belonging to a cluster including G mutation S100G. This cluster was dominant both in our samples and in the European samples overall. The clinical impact of the circulating variants and their mutational profile was not studied for this publication.

It must be noted that the phylogenetic analysis included samples that came from one single city in Spain (Vigo) and the GISAID database. The European GISAID sequences came mostly from England. This causes an incomplete picture of RSV in Europe. An increase in available sequences would provide deeper insights into the circulation of RSV. Additionally, although this study focused on European samples due to geographical proximity to our samples, future analysis could include a phylogenetic analysis of all global sequences.

There are limitations to our study. Our final protocol was only tested on 15 samples, which ranged in low-to-mid CT value (14–23). Therefore, samples with a low viral load were not included. Our data suggest that viral load has an impact on coverage depth and, therefore, samples with a higher CT value could provide lower yield. Additional pretreatment steps could be added to the protocol to improve sequencing yield in samples with a low viral load, such as viral isolation, viral culture, or DNAse treatment. In a recent publication, Dong et al. showed that a DNAse treatment prior to an RT-PCR improved the yield of sequencing by removing host DNA [17]. As a second limitation, samples were not selected from the sentinel surveillance system and therefore there could be a bias introduced in the individuals selected. The majority of the samples came from the pediatric emergency service of the CHUVI (80%), suggesting a potential bias on disease severity for the samples sequenced. Finally, we only tested samples from our facility. In order to ensure the quality of the protocol, samples from different locations should be tested.

Despite the limitations in this process, a simplified protocol for enrichment such as this one would make RSV-A and B whole-genome sequencing more accessible to epidemiology services throughout the world. An increased number of sequences available would enhance epidemiological vigilance and put more alertness on mutations causing resistance to available treatments and prophylaxis. It would also make it easier to update the primer panel for each epidemiological season, as all circulating variants would be accounted for.

## 4. Materials and Methods

### 4.1. RSV Clinical Isolates

Nasopharyngeal swabs from patients with a suspected viral respiratory infection were processed in the Microbiology Department of the Complexo Hospitalario Universitario de Vigo. The diagnostic and semiquantification of viral load via real-time RT-PCR (CT value) were performed using the kit Seegene AllPlex Respiratory panel 1 (Seegene Inc., Seoul, Republic of Korea).

From October 2022 to March 2023, 135 viral genomes were selected for sequencing through the optimization steps of this panel, while an additional 15 samples were used to test the final version of the protocol (8 RSV-A and 7 RSV-B). All samples studied had a CT value between 14 and 23. All sequences are available in the GISAID database (Appendix A). The classification of the samples used during this study is shown in Appendix A.

No ethical approval was required for this study as the authors had no access to patient-identifying information and they were not part of the data collection. 

### 4.2. RSV-Targeted Amplification and NGS

RNA extraction from RSV-positive samples was performed (QIASymphony DSP Virus/Pathogen Midi Kit, Qiagen, Hilden, Germany). Our design of primers was built upon a previously published panel [16]. In summary, 39 primers divided into two pools were used (Appendix A), with each pool covering half of the genome. For retrotranscription, amplification, and library generation, Illumina CovidSeq was used (Illumina, San Diego, CA, USA). The mix of reagents for the RT-PCR was performed based on a protocol published for the amplification of Influenza A and B genomes [30]. Each reaction included 15 μL of IPM, 3.2 μL of FSM, 1.2 μL of the primer mix at 10 μM concentration, and 3.6 μL of nuclease-free water and 1 μL of RVT. Reagent names are the commercial names of the reagents in the Illumina CovidSeq kit. An RT-PCR was performed in the following manner: 42 °C for an hour, 98° 2′, and 35 cycles of 98° 15 s and 63 °C 7 min. Two reactions per sample were required. The detailed protocol is available online [31]. As per the CovidSeq reference guide, 10 μL of each reaction was mixed (one per primer pool), and then the same procedure as in the kit was performed for library generation. Amplicons and libraries were quantified using QubitFlex (Invitrogen, Eugene, CA, USA) and checked on an Agilent Bioanalyzer 2100 (Agilent Technologies, Santa Clara, CA, USA). Libraries pooled according to protocol were sequenced on an Illumina iSeq 2 × 151 platform, using 100 pM as the final concentration and spiked with 1% PhiX. (Illumina, San Diego, CA, USA). One negative control (Non-Template Control) was added and then sequenced.

### 4.3. Data Analysis

The quality of the fastq files was checked (FastQC 0.11.9, QualiMap 2.2.1) [32,33]. For each sample, reads were aligned to the references MN078114.1 and ON729320.1 (BBMap version 39.01) [34]. Reads were merged with SAMtools mpileup and removed (iVar 1.3) [35] if below 32 bp long or a 20 quality threshold. Consensus sequences were generated (minimum read depth of 10, iVar 1.3). All variants with a minimum VAF threshold of 0.01 were registered. Genome coverage was calculated (SAMtools v1.10, htslib 1.10.2) [36]. Sequences showing a genome coverage <80% along the whole genome or in the genomic area around the second hypervariable region of the G gene were removed. Finally, the RSV genotype was assigned (Nextclade) [37] (Figure 5).

Output consensus sequences were aligned using MAFFT. MEGA11 was used for the construction of phylogenetic Maximum-Likelihood trees (Jukes–Cantor Method). Presentation and visualization of trees was performed using Microreact. Additional phylogenetic analysis was performed using Ultrafast Sample placement on Existing tRee (UShER) [38]. Base numbers in the RSV genomes were indicated using Nextclade references hRSV/A/England/397/2017 and hRSV/B/Australia/VIC-RCH056/2019 for RSV-A and RSV-B, respectively.

All plots were designed in R using the ggplot R package [39]. Genetic distance between samples was performed using the DistanceMatrix function from R, which calculates the Hamming distance between sequences. Gap–gap matches and terminal gaps were excluded from the distance calculation.

## Figures and Tables

**Figure 1 ijms-24-16055-f001:**
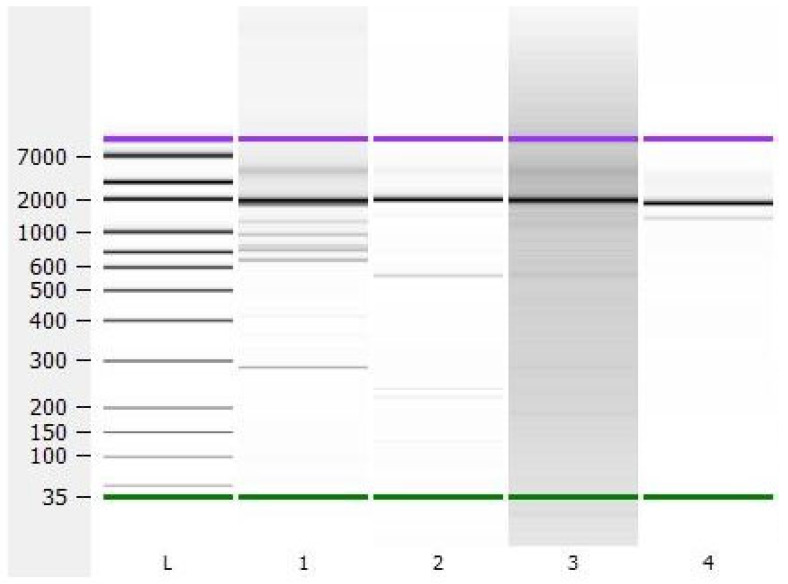
A representative image of an Agilent bioanalyzer for the amplification products. Channels from left to right: ladder (L), RSV-A, pool 1 (1); RSV-A, pool 2 (2); RSV-B, pool 1 (3); RSV-B, pool 2 (4). Scale indicates size in base pairs. Green = light marker, 35 base pairs. Purple = heavy marker, 10,380 base pairs.

**Figure 2 ijms-24-16055-f002:**
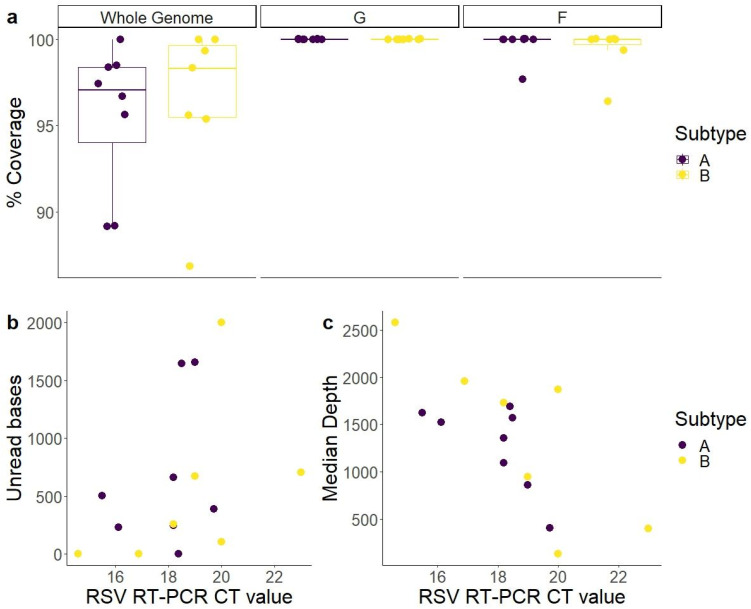
Successful amplification and sequencing of RSV samples. (**a**) Coverage (base reads > 10) for the whole genome (left), G gene (center), and F gene (right). All samples had a genomic coverage above 85%. (**b**) Correlation between the RSV CT value of the original sample and the number of bases with a coverage below 10 (unread). (**c**) Correlation between the RSV CT value and median base depth (reads per base). The CT value only had an impact on median depth (R^2^ = 0.06 and 0.49, respectively). The CT value was obtained using real-time RT-PCR with the Seegene AllPlex Respiratory panel 1 kit.

**Figure 3 ijms-24-16055-f003:**
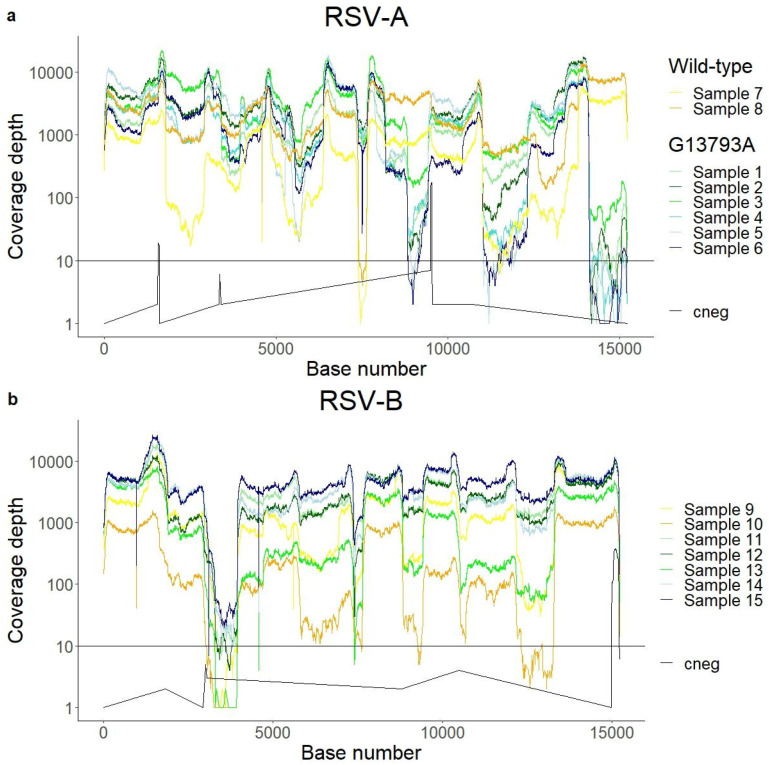
The coverage profile across the genome of RSV-A (**a**) and RSV-B (**b**). The drop in reads in amplicon 10 of RSV-A is shown to be cluster-specific, with six out of eight samples showing a drop in reads. The negative control (Non-Template Control), aligned against the RSV-A and RSV-B references, is shown in black.

**Figure 4 ijms-24-16055-f004:**
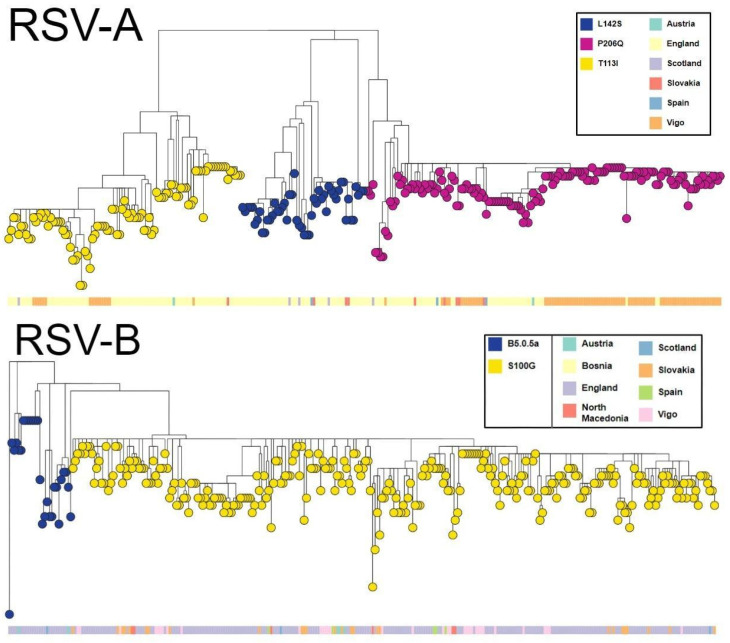
Phylogenetic trees with samples from this study and European samples in the GISAID database from October 2022 to March 2023. RSV-A (top) and RSV-B (bottom). For RSV-A, all samples were added (*N* = 195). For RSV-B, 300 out of a total of 492 were randomly selected to facilitate visualization. Tips are colored by cluster, with each cluster defined by a mutation in the gene of the G protein. At the bottom of each tree, samples are labeled according to their country of origin. Samples from Vigo are those sequenced for this study. The trees were generated with microreact. Microreact files can be found in the Appendix A.

**Figure 5 ijms-24-16055-f005:**
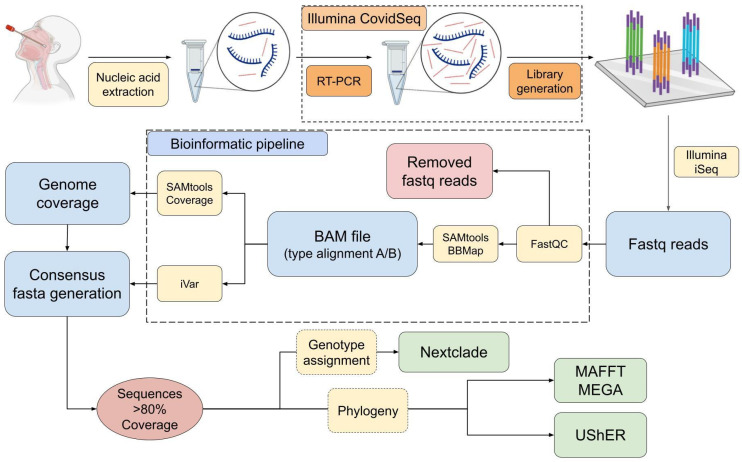
Scheme of RSV molecular surveillance based on whole-genome sequencing. The process from the clinical sample to the assignment of RSV genotype, mutation profile, and phylogeny is shown. Figure partially generated with BioRender.

**Table 1 ijms-24-16055-t001:** Mutations detected in the primer binding areas from the original primer panel used for this publication. The mutation frequency in global sequences is included, based on sequences uploaded to GISAID between October 2022 and March 2023. Low-coverage sequences were excluded from the search. Base numbers were according to reference sequences hRSV/A/England/397/2017 and hRSV/B/Australia/VIC-RCH056/2019.

Mutation	Primer Affected	Frequency (2022–2023)
C7933T	A5r	31%
C7939T	A5r	12%
G13793A	A10f	46%
A12183T	B8r	85%

**Table 2 ijms-24-16055-t002:** Characteristics of the RSV-infected individuals for the sequences in this study.

		*N*	%
Subtype	RSV-A	95	64.6
	RSV-B	49	33.3
	Coinfection	3	2.0
Gender	Female	72	49.0
	Male	75	51.0
Age	<10	124	84.4
	>50	23	15.6
Total		147	100

## Data Availability

Sequences generated for this publication are uploaded to the GISAID database. Accession ID numbers for all sequences can be found in Appendix A. The detailed amplification protocol for RSV can be found in protocols.io: (dx.doi.org/10.17504/protocols.io.eq2lyjzbrlx9/v2) Accessed on 1 November 2023.

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
