# Peer review of "The Modification of the Illumina® CovidSeq™ Workflow for RSV Genomic Surveillance: The Genetic Variability of RSV during the 2022–2023 Season in Northwest Spain"

_ijms, 2023, doi:10.3390/ijms242216055_

Round 1

Reviewer 1 Report

Comments and Suggestions for Authors

In this manuscript “Modification of the Illumina® CovidSeq™ workflow for RSV 2 genomic surveillance: Genetic variability of RSV during 3 2022-2023 season in northwest Spain”, Carlos Davina-Nunez et al. developed a new method to improve surveillance for both RSV-A and B. There procedures allow also to highlight mutations of interest that could decrease the efficacy of antivirals and monoclonal antibodies. The fact that they can amplify simultaneously RSV-A and B relatively easily and at a reasonable cost is very interesting. I only have minor comments :

1) Maybe the authors could add a bit more background on RSV for those who are not expert on this virus (order, familly, number of proteins, size of the genome etc.)

2) Maybe the GISAID sequences will be available later or I failed to access this data but for some reason I was not able to find them with the given access numbers.

3) Figure 1 : Do the authors have some comments regarding the sample 1 that seems less clean than the others?

4) Figure 2 : The method  used in this figure is not that clear. Maybe the authors could explain a bit more how they obtain these CT.

5) Figure 4 : The trees are very interesting, however it would help to have better quality images or even better : an access to the interactive version on microreact if this is possible for publication.

Author Response

Pleas see the attachment

Reviewer 2 Report

Comments and Suggestions for Authors

The author well presented detection of arising mutations based on RSV genomic surveillance with modified protocol.But, i have some minor comments on manuscript.

1. In phylogenetic tree, the author donot include Asia strains. Please discuss on it.

2. CT value means real time PCR CT value, isnot it? It is better to describe clearly in materials and methods. Please write full name of CT.

3. Based on your data, why CT value >23 samples didnot include your study? Please discuss it.

4. Did the author try virus isolation on those positive RSV samples? please discuss it.

5. A total of 135 viral genome positive samples, why the author did 15 samples only used or analysis?Please discuss it.

6. Regarding 4 mutation in Table 1, those mutations are related with clinical severity or not?

7. Did the author tried modification protocol in other respiratory viruses (eg-SARS-COV-2)?
